# Involvement of Oxytocin and Progesterone Receptor Expression in the Etiology of Canine Uterine Inertia

**DOI:** 10.3390/ijms232113601

**Published:** 2022-11-06

**Authors:** Carolin Jungmann, Caroline Gauguin Houghton, Frederik Goth Nielsen, Eva-Maria Packeiser, Hanna Körber, Iris M. Reichler, Orsolya Balogh, Sandra Goericke-Pesch

**Affiliations:** 1Reproductive Unit, Clinic for Small Animals, University of Veterinary Medicine Hannover, 30559 Hanover, Germany; 2Section for Veterinary Reproduction and Obstetrics, Department of Clinical Veterinary Sciences, University of Copenhagen, 1870 Frederiksberg, Denmark; 3Clinic of Reproductive Medicine, Vetsuisse Faculty, University of Zurich, 8057 Zurich, Switzerland; 4Department of Small Animal Clinical Sciences, Virginia-Maryland College of Veterinary Medicine, Blacksburg, VA 24061, USA

**Keywords:** dog, dystocia, uterine inertia, oxytocin receptor, progesterone receptor

## Abstract

An altered oxytocin and progesterone receptor (*OXTR* and *PGR*, respectively) expression was postulated in canine uterine inertia (UI), which is the lack of functional myometrial contractions. *OXTR* and *PGR* expressions were compared in uterine tissue obtained during C-section due to primary UI (PUI; *n* = 12) and obstructive dystocia (OD, *n* = 8). In PUI, the influence of litter size was studied (small/normal/large litter: PUI-S/N/L: *n* = 5/4/3). Staining intensity in immunohistochemistry was scored for the longitudinal and circular myometrial layer and summarized per dog (IP-Myoscore). Mean P4 did not differ significantly between PUI (*n* = 9) and OD (*n* = 7). *OXTR* and *PGR* expressions (ratios) were significantly higher in PUI (*OXTR*: *p* = 0.0019; *PGR*: *p* = 0.0339), also for *OXTR* in PUI-N versus OD (*p* = 0.0034). A trend for a higher *PGR* IP-Myoscore was identified (PUI-N vs. OD, *p* = 0.0626) as well as an influence of litter size (lowest *PGR*-Myoscore in PUI-L, *p* = 0.0391). In conclusion, PUI was not related to higher P4, but potentially increased *PGR* availability compared to OD. It remains to be clarified whether *OXTR* is upregulated in PUI due to a counterregulatory mechanism to overcome myometrial quiescence or downregulated in OD due to physiological slow *OXTR* desensitization associated with an advanced duration of labor. Identified *OXTR* differences between myometrial layers indicate the need for further research.

## 1. Introduction

Uterine inertia (UI) is, with approximately 75%, the most common maternal cause of dystocia in the pregnant bitch [1,2]. Regardless of the varying criteria applied by the different authors [1,3,4], UI is generally divided into primary and secondary UI [2]. Primary uterine inertia (PUI) is characterized by the absence of functional myometrial contractions preventing natural delivery despite normal-sized fetuses and a patent birth canal. Secondary uterine inertia (SUI) is caused by myometrial exhaustion due to obstruction, usually after one or several pups have already been expelled [1,2,3,5]. As uterine inertia possesses a high risk for maternal health and fetal survival, early recognition and rapid treatment are paramount [1,6]. However, conservative medical treatment, consisting of oxytocin, calcium, dextrose, or denaverine alone or in combination [1,6,7], often remains unsuccessful and, thus, an emergency Cesarean section (C-section) is required in over 60% of canine dystocia cases [1,6,8,9]. As a result, C-sections are frequently considered as the first choice without any prior medical treatment attempts [7,10].

Despite its high incidence, the etiology of PUI is poorly understood so far. Nevertheless, predispositions have been described for several breeds [3,11,12,13] as well as risk factors, such as maternal overweight, advanced age, very small or large litter size, and hormonal, electrolyte, or metabolic imbalances [1,2,3,6,10,12,14,15]. Additionally, alterations in the expressions of basic contractile and contractility associated proteins in the uterus, e.g., smooth muscle γ-actin, smooth muscle myosin [16], the RhoA/Rho associated kinase pathway, and leptin signaling [17], were found in cases of PUI.

Parturition is a delicately orchestrated endocrine event with progesterone (P4), prostaglandins, and oxytocin playing key roles, not only in the dog [18,19]. Different studies have looked deeper into peripheral hormone concentrations during canine parturition in general and specifically in UI [14,15,20,21,22]. The onset of labor is inseparable from luteolysis and, thus, P4 withdrawal, as high plasma P4 levels mediate myometrial quiescence and prevent uterine contractions [23,24]. In contrast to luteal regression in nonpregnant bitches, the slow regression is accelerated around day 60 of pregnancy, when reaching a lower threshold level activates the prepartum luteolytic cascade [25,26,27]. The following steep decline in P4 is accompanied by a large increase in PGF2α-metabolite (PGFM) concentrations [22,26], indicating active luteolysis. Feto-maternal communication seems to play a crucial role in activating the prostaglandin system and, thus, the prepartum PGF2α release (reviewed in [28]). Interestingly, a significantly higher P4:PGFM ratio in dystocic bitches supporting the postulated failure of luteolysis in PUI [20,29,30,31] was found in one study [20], but not in another [32], emphasizing the complexity of UI. In addition to P4, expression of the P4 receptor (*PGR*) likely plays a pivotal role in pregnancy and parturition as indicated by decreased *PGR* expression in uterine tissue close to term (day 60 after the LH surge) [33]. Although the hormonal mechanisms behind the regulation and expression of *PGR* in the corpus luteum and placenta in relation to parturition have already been investigated for the dog [27,34,35,36,37], research on (myometrial) *PGR* expression, especially related to dystocia and PUI, is still lacking.

Another potent uterotonic hormone crucially required for parturition and expulsion of the fetuses is oxytocin (OXT). OXT enhances the contractility of the myometrium through binding to its G protein coupled receptor (*OXTR*), activating the protein kinase type C (PKC) pathway [38]. During birth, OXT is released from the neurohypophysis as a response to intracervical pressure, activating uterine contractions (Ferguson’s-reflex) [39]. As OXT sensitivity is significantly increased around canine parturition due to upregulation of *OXTR* expression [33,40,41], low OXT concentrations [10,42] or abnormal *OXTR* expression could contribute to the occurrence of PUI. Furthermore, in cattle [43] and sheep [44], OXT was shown to affect the release of prostaglandins, primarily PGF2α, and to induce the expression of the uterine prostaglandin-endoperoxide synthase 2 (PTGS2), additionally stimulating uterine contractility indirectly. Although *OXTR* might play a significant role in canine UI, information about its expression in dystocic bitches is limited to the mRNA level [32], not allowing for a final conclusion about its role in canine UI.

Summarizing the current state of knowledge, it is obvious that the role of P4 and OXT and their receptors in canine parturition and canine UI require further elucidation, not only for a deeper understanding of the physiology and pathophysiology, but also to optimize medical treatment options and to develop possible preventive strategies. Consequently, this study investigated the expression of *PGR* and OXT in bitches diagnosed with PUI and OD, hypothesizing altered expressions in canine PUI.

## 2. Results

### 2.1. Determination of Serum P4 Concentrations

P4 concentrations ranged between 1.3 and 4.6 ng/mL independent of groups. The unpaired *t*-test revealed no significant differences when comparing PUI and OD (Figure 1).

### 2.2. OXTR Expression

*OXTR* IP mRNA expression (ratio) differed significantly between PUI and OD (*p* = 0.0019) with a higher expression in PUI (Figure 2a). Similarly, *OXTR* IP gene expression was significantly increased in PUI-N compared to OD (*p* = 0.0034) (Figure 2b). In contrast, no effect of litter size on *OXTR* IP mRNA expression was found within the PUI group (Figure 2d). Likewise, a comparison of *OXTR* expression between PUI and OD at UP or between IP and UP (PUI and OD summarized) revealed no significant differences (Figure 2c,e).

Specific staining for *OXTR* was observed in the perinuclear area of the myocytes in both myometrial layers and, less intensely, in endometrial stromal cells (Figure 3). In addition, maternal decidual cells and capillary pericytes stained positively for *OXTR* in the placenta.

No significant differences were identified in terms of *OXTR* staining between dystocia groups, neither individually in the longitudinal and circular myometrial layer nor in the summarized IP-Myoscore (Appendix A). In addition, *OXTR* staining was not influenced by litter size (Appendix A). However, independent of groups (PUI, OD, and PUI + OD), *OXTR* staining intensity was significantly stronger in the circular compared to the longitudinal layer in IP with the difference becoming even more obvious when combining both localizations (IP + UP) (*p* < 0.001; Figure 4; Appendix A). Appendix A contain all statistical analysis regarding differences in *OXTR* staining intensity between myometrial layers and groups.

### 2.3. PGR Expression

In IP, a significantly higher *PGR* mRNA expression (ratio) was observed in PUI compared to OD (*p* = 0.0339), with a similar trend identified when comparing PUI-N and OD (*p* = 0.0524) (Figure 4). This effect was not observed in UP samples. In addition, litter size did not have an impact on *PGR* expression, and neither were differences identified between IP and UP (Figure 5d,e).

In IHC, myometrial smooth muscle cells stained strongly positive for *PGR*. Additionally, a specific immunopositive signal was detected in endometrial stromal cells, uterine glandular cells, and luminal epithelial cells. In the placenta, only maternal decidual cells stained *PGR*-positive (Figure 6).

Statistical analysis of *PGR* IHC results is given in the Appendix A. The *PGR* staining score did not differ between the longitudinal and circular myometrial layer (Appendix A). Comparison of *PGR* IP-Myoscore from PUI and OD samples did not reveal significant differences but a tendency of higher staining intensity in PUI that was also evident for PUI-N versus OD (Figure 7, Appendix A). Litter size influenced *PGR* expression (*p* = 0.0391), with the lowest score in PUI-L compared to PUI-N/S. (Figure 7). This effect was also apparent when comparing the results of the longitudinal myometrial layer (Kruskal–Wallis, *p* = 0.0306), but not of the circular layer (Appendix A).

## 3. Discussion

Despite the frequent occurrence of PUI in dogs and its severe consequences for maternal and offspring survival, the exact etiology is still unknown. Up to now, only few studies have investigated its pathophysiology from a molecular biology approach looking directly at the level of the uterus [16,17,32,45,46]. As both oxytocin and progesterone are crucial hormones during parturition, we hypothesized and identified altered expression of the respective receptors as potential causes for PUI.

The expression of the *OXTR* at the mRNA and protein level, in relation to pregnancy and normal, undisturbed parturition, has already been extensively studied [32,40,41]. However, by now, only Tamminen et al. [32] investigated a possible role of *OXTR* mRNA expression in the context of dystocia in dogs, using full-thickness uterine tissues obtained from the incision site during C-sections. They identified the lowest *OXTR* expression in OD bitches, with the expression being significantly lower than in bitches presented for elective C-section because of small litter size before term or previous dystocia (ECS). Although *OXTR* expression was lower in OD compared to complete uterine inertia (CUI), the difference was not significant [32]. It remains to be clarified whether the identified *OXTR* differences were related to the smaller litter size (singleton or two puppies) in some ECS bitches or to the differences in the endocrine situation of prepartum (ECS) and intrapartum (OD) bitches, as parturition is a sensitive orchestrated endocrine event with significant changes occurring within the last 24 h before initiation of labor [24,47,48]. Samples from ECS bitches likely do not represent the physiological situation (including gene and protein expression) during canine parturition and this is a—however ethically easy to understand—limitation of this and our own study, too, that no uterine tissue samples of eutocic bitches were included for comparison.

Nevertheless, Tamminen et al. [32] suggested that *OXTR* was downregulated in OD due to prolonged action of OXT, leading to myometrial exhaustion. In human myometrium, a physiological slow *OXTR* desensitization, accompanied by *OXTR* mRNA downregulation, proceeds with an advanced duration of labor [49]. Assuming a homologous event in dogs as described before in women, decreased *OXTR* mRNA expression in OD, as also observed in our study, might be caused by a prolonged labor itself or altered (accelerated in this case) *OXTR* desensitization. Furthermore, prolonged intracervical pressure, resulting in increased hypophyseal OXT release, could contribute to this desensitization. A decreased mRNA expression of other contractile and contractility associated proteins in IP sections of OD compared to PUI bitches has also been shown in our previous reports [16,17], corroborating the current findings and the proposed role of prolonged labor contractions on the uterus. Indeed, all OD bitches included in our study still showed strong straining as a response to digital vaginal feathering and spontaneous abdominal and/or uterine contractions, indicating that these bitches did not reach myometrial fatigue, causing secondary UI. Although pregnancy regulation in the dog has not been fully elucidated until now, normal parturition is associated with a rapid progesterone decline in response to increased luteolytic concentrations of prostaglandins, mediated by placental feto-maternal crosstalk, allowing strong aligned uterine contractions [27,40,50]. The significantly higher *OXTR* mRNA expression (ratio) in IP samples obtained from PUI (and PUI-N) dams compared to OD might be a consequence of the absence of intracervical pressure due to the fact that no puppy had been born and, thus, having insufficient systemic OXT release, as previously postulated by Tamminen et al. [32]. It is worth noting that comparison of the OD and PUI samples was shown to be suitable in previous studies to gain further insight into the pathophysiology of canine dystocia and specifically PUI [15,16,17,32,45,46].

At the protein level, localization of *OXTR* expression was in accordance with other authors [40,51,52]. They described a strong immunopositive signal in the uterine surface epithelium, superficial uterine glands, vascular endothelial cells, and stroma cells of the endometrium in early pregnancy that becomes noticeably weaker during prepartum luteolysis [40]. Similar to our results, a strong signal was localized in placental decidual cells and capillary pericytes and the myocytes of both myometrial layers [40,51]. Thus, our findings confirm the previously described colocalization with *PGR* in the placenta. This suggests that *OXTR* is involved in the signaling cascade, possibly resulting in an increased prepartum output of luteolytic PGF2α by blocking *PGR* function [40,50].

The significant differences in *OXTR* expression between groups obtained at the mRNA level using full-thickness uterine tissues were not confirmed when considering myometrial staining only as identified by IHC. Various explanations for this phenomenon are possible including limited suitability of IHC for protein quantification, subjectivity of the investigators, and slight changes in the environmental settings [53]. As Gram et al. [40] identified the highest *OXTR* expression in the myometrium compared to endometrium and placenta when investigating uteroplacental compartmentalization, it seems appropriate to focus on myometrial *OXTR* protein expression. Moreover, the observed disparity of results might be due to a difference between the possible transcription of *OXTR* mRNA and actual translation and, thus, expression of the receptor at the protein level. Likewise, the increased *OXTR* mRNA expression in PUI might induce a subsequent upregulation of the *OXTR* protein expression, possibly to increase responsiveness in the case of insufficient OXT release/availability. Low OXT concentrations in the peripheral blood had been identified earlier and postulated to be involved in the etiology of PUI in a cohort of dogs [20]. Consequently, an increased *OXTR* availability might be a counterregulatory mechanism to increase responsiveness to low OXT levels to successfully expel puppies. Although determination of peripheral blood OXT concentrations would have provided valuable information, it was not included in this study, due to well-known difficulties in analysis because of the short half-life [54], its strong binding to other molecules, and possible tests’ cross-reactivity [31].

Interestingly, significant differences in *OXTR* protein expression between the two myometrial layers were identified in IP samples. Evidence for distinctions in function, morphology, and innervation between myometrial layers have already been described for rats [55], rabbits [56], pigs [57], and cattle [58,59]. An ex vivo organ bath study, investigating the contraction behavior of canine periparturient and parturient myometrium, performed by our working group, supports the current findings of *OXTR* expression being significantly increased in the circular layer [60]. In these experiments, the circular myometrial layer visually showed a better response to oxytocin stimulation compared to the longitudinal layer, especially when using higher OXT concentrations [60]. In contrast to this, Gogny et al. [61] described a greater contractile response to OXT in the longitudinal compared to the circular myometrial layer of cyclic healthy bitches in anestrus or metestrus with and without Aglepristone treatment. Recently, our working group investigating the role of prostaglandins in PUI identified a significant difference in PTGS2 staining between myometrial layers, with PTGS2 being higher-expressed in the longitudinal layer of the same samples [46]. Taken together, all studies suggest a complex regulation of myometrial contractile activity during parturition with coordinated action of the two myometrial layers in canine uterine tissues being essential for eutocia. Although some studies have investigated functions of myometrial contractions in delivering the offspring in pigs [62,63,64], mice [65], and cattle [66], little is known about the mechanisms in the dog. It can be assumed that birth mechanisms in the dog, another polytocous species, are similar to the porcine mechanisms with caudally (toward the cervix) and cranially (away from the cervix) directed contractions moving the piglets forward and backward within the uterine horn during the expulsive stage [64]. Similar alternating patterns of fetal expulsion from collateral horns were described for both species [63,67,68]. However, the significance of the two myometrial layers and their postulated different functions in this context urgently requires further investigation. Differentiating the longitudinal and circular myometrial layer individually for layer-specific analysis of *OXTR* gene expression by qPCR and *OXTR* protein quantification by Western blotting could further support our results and hypothesis.

Another endocrine event of utmost importance for canine parturition and functional myometrial contractions due to sensitization to OXT is the prepartum progesterone drop [41]. Failure of luteolysis and, thus, higher serum P4 concentrations have been discussed to be related to PUI [2,20]. In addition to P4 itself, an important role of *PGR* for (normal) parturition was postulated in the dog [28,50]. For this reason, P4 and *PGR* in canine PUI and OD patients were analyzed, without identifying a significant difference, however. Interestingly, and other than expected and previously postulated [20,29,30,31], we identified no significant differences between P4 serum concentrations in PUI compared to OD, clearly showing that, at least in our cohort, PUI was not due to inadequate luteolysis. To gain further insights into the role of *PGR* for canine parturition and based on the hypothesis that *PGR* expression is altered in PUI, our working group was the first investigating *PGR* mRNA and protein expression in relation to canine dystocia. *PGR* mRNA expression was significantly higher in PUI compared to OD in IP samples, with the same trend observed for PUI-N. These findings suggest increased sensitivity of the uterus even in cases of low circulating P4 concentrations, possibly resulting in a higher degree of inhibition of myometrial contractions in PUI compared to OD, contributing to the failure of contractions. Two *PGR* isoforms, A and B, exist due to the use of alternative translation initiation sites. Peavey et al. [69] identified opposing actions of the two isoforms on myometrial contractility in mice. Whereas PGR-B overexpression significantly increased gestation length and hampered uterine contractility as demonstrated by a significantly lower contractile response of the myometrium in an ex vivo dose–response experiment, excessive PGR-A expression induced an intensified contractile response to uterotonic agents [69]. Thus, overexpression of PGR-B resulted in a relaxation of the myometrium and reduced *OXTR* expression, hindering the myometrium to contract sufficiently when oxytocin was administered. Assuming that in dogs, a similar effect of *PGR* isoforms exists, increased PGR-B expression in PUI could contribute to the reduced ability to functionally contract. However, as no information about *PGR* isoform expression is currently available for the dog, except for the mammary gland [70], and the primers and antibody used in this study do not discriminate between the isoforms, the *PGR* isoforms and their expressions deserve further investigation in the canine uterus, especially, but not only in case of dystocia.

*PGR* intracellular localization was as previously described in pregnant dogs, with endometrial stromal cells, myometrial smooth muscle cells, and maternal decidual cells staining strongly [71,72]. Although the significant difference between PUI and OD identified in *PGR* mRNA expression was not confirmed on the protein level, a trend for a higher IP-Myoscore in PUI/PUI-N compared to OD was found. Altered tissue composition (full-thickness uterine tissue for mRNA analysis versus myometrium only for protein IHC scoring) might justify the identification of a trend only. However, local changes in terms of *PGR* expression related to PUI have to be considered. It remains to be clarified whether this altered *PGR* expression is involved in the development of PUI or is a result or consequence of inadequate or absent endocrine or mechanical stimuli due to PUI. Unlike the findings for *OXTR*, *PGR* expression in the two myometrial layers did not differ. Considering the function of P4 and *PGR*, namely maintaining pregnancy, it seems evident that inhibition needs to be equal in both layers to successfully prevent contractions before term. Thus, the equal *PGR* distribution in both layers seems to represent a physiological situation.

Litter size is known to be a potential risk factor for development of PUI in bitches [12] with increased PUI risk described in the case of small (namely singleton) or large litters. Analysis of *PGR* protein expression related to litter size revealed a significantly lower myometrial *PGR* expression in the PUI-L group. Myometrial overstretching due to a large litter was already mentioned earlier in relation to UI [3,5,11]. Overstretching could reduce the density of *PGR*s per cm^2^, explaining the reduced *PGR* expression in PUI-L. However, this observation has to be treated with caution, as the PUI-L group is very small (*n* = 3). Interestingly, no effect of litter size was observed for *OXTR* in this study, while in the same bitches, an effect on the expression of smooth muscle γ-actin, a basic contractile protein, was found [16]. In conclusion, further studies are necessary to confirm if the etiology and pathophysiology of PUI differ depending on the litter size.

The relatively small sample size might display a limitation of the current study, as proving significance is complicated when comparing smaller groups, such as the PUI-N group. Heterogeneity of the bitches in terms of age, breed, and body weight represents the canine population [46], but might have influenced the present results, too. However, detailed examinations of bitches before surgery, complete medical history, and strict adhesion to inclusion criteria are a large benefit of our studies. In conclusion, to gain more precise insights into canine uterine inertia and possibly dystocia, investigations on a larger population including IP and UP samples should be conducted, with the difficulties in obtaining UP samples (by spaying at the time of C-section) being previously discussed in detail [46]. In addition, breed-specific studies might be considered for comparative use, too.

## 4. Materials and Methods

### 4.1. Animals and Study Design

The samples utilized in this study were already used previously by our research group [45,46] and our collaborators [15,16,17].

Full-thickness uterine tissue samples were collected from 20 bitches at term presented for medically indicated emergency C-sections following owner’s consent. Apart from dystocia, the bitches were clinically healthy and received no ecbolic or tocolytic medication before the samples were taken. Detailed general, medical, and reproductive history of all dogs was recorded. The mean age of the bitches included was 4.2 ± 2.1 years, and the mean body weight was 20.1 ± 18.4 kg. All bitches had a thorough physical and obstetrical exam. Fetal heart rates were monitored by abdominal sonography. Additionally, when required for a proper diagnosis and it was feasible in terms of the dam’s and fetal health, radiographs were taken and tocodynamometry was performed. Blood samples were collected pre-surgery to obtain baseline hematology and chemistry including ionized calcium and glucose [15]. In addition, blood was sampled from 16 bitches for retrospective analysis of serum progesterone (P4) concentrations using radioimmunoassay (RIA) [22,25,73].

### 4.2. Grouping

The grouping of the bitches was as previously described [46]. All inclusion criteria are given below. The bitches were retrospectively assigned to one of the subsequent groups: 1. primary uterine inertia (PUI, *n* = 12), or 2. obstructive dystocia (OD, *n* = 8).

Bitches included in the PUI group had failed to deliver any puppy at term, and showed a lack of the Ferguson’s reflex at vaginal stimulation and only weak or no abdominal contractions. Obstruction was ruled out by vaginal examination and X-rays. In addition, the bitches had to show one of the following signs: The prepartum temperature drop was more than 20 h ago or/and signs of first stage labor were shown for ≥20 h or the temperature had already normalized without progression to second-stage labor; no signs of second-stage labor, but green vulvar discharge for >2 h; unproductive, weak, infrequent abdominal contractions for >4 h without progression; or no abdominal contractions, although fetal fluids passed more than 3 h ago. As the litter size was quite variable, the PUI group was subdivided into small/normal/large litter size (PUI-S, PUI-N, PUI-L) in relation to the average litter size of the respective breed [74]. Litter size within the breed average ± 1 standard deviation (SD) was considered as normal (PUI-N), whereas less or more than ± 1 SD was assigned to PUI-S or PUI-L, respectively [15,16,17,45,46].

Bitches diagnosed with OD were used for comparison with PUI. These bitches were in second-stage labor but failed to deliver due to obstruction of the birth canal, as confirmed by digital vaginal palpation and/or abdominal X-ray. However, they were still showing strong abdominal contractions, either spontaneously or in response to feathering (digital vaginal stimulation). These obviously functional contractions provided the basis for our assumption that OD is suitable for comparison with PUI. The litter size of all OD bitches was normal compared to the described average litter size of the breed [74].

Bitches with signs of a systemic illness or other risk factors for UI were excluded from the study.

### 4.3. Tissue Sample Collection and Processing

During medically indicated C-section, uterine tissue samples were obtained after all puppies had been delivered. A full-thickness tissue biopsy, approximately 0.5–1 cm wide, was taken from the interplacental (IP) tissue (between two placentation sites) along the uterine incision line. If a concomitant ovariohysterectomy (Sectio Porro) was medically indicated or requested by the owner, IP samples were collected after extraction of the uterus and additional tissue stripes from utero-placental sites (UP) and the placenta were taken.

Collected tissue was divided into different parts for subsequent mRNA and protein analysis, as well as for histology and immunohistochemistry (IHC) as previously described [45,46]. For mRNA analysis, samples were covered with RNAlater^®^ (Ambion Biotechnologie GmbH, Wiesbaden, Germany) immediately post-surgery, temporarily stored at 4 °C, before being kept at −80 °C until extraction. For histology and IHC, tissue samples were fixed in 10% neutral phosphate-buffered formalin for 24 h at 4 °C, washed regularly with phosphate-buffered saline for several weeks, and subsequently embedded in paraffin.

### 4.4. RNA Isolation and Reverse Transcription

RNA isolation using a Trizol (SIGMA-ALDRICH CHEMIE GmbH, Steinheim, Germany)-based protocol was performed as previously described [45,46]. To evaluate and measure the extracted mRNA concentrations, an IMPLEN NanoPhotometer^®^ (IMPLEN GmbH, Munich, Germany) was used.

Reverse transcription for full-length first-strand cDNA was performed with DNase-pretreated RNA (200 ng/µL) and the RevertAidFirst Strand cDNA synthesis Kit (#K1622, Thermo Scientific, Waltham, MA, USA) according to the manufacturer’s instructions and as previously described [45,46]. cDNA was stored at −20 °C until further use. Specific primer sets for both *OXTR* and *PGR* for quantitative real-time polymerase chain reaction (RT-qPCR) were purchased from TAG Copenhagen (TAG Copenhagen A/S, Copenhagen, Denmark) and primers for the reference genes *glyceraldehyde-3-phosphate dehydrogenase* (*GAPDH*), *protein tyrosine kinase 2* (*PTK2*), *eukaryotic translation initiation factor 4H* (*EIF4H*), and *lysine-specific demethylase 4A* (*KDM4A*) [75] from Microsynth (Microsynth AG; Balgach, Switzerland) (Table 1). The specificity of the primers was checked with BLAST (http://blast.ncbi.nlm.nih.gov, accessed on 3 August 2021).

For RT-qPCR, a LightCycler^®^ 96 real-time PCR system (Software version 1.1.0.1320, Roche Diagnostics GmbH, Mannheim, Germany) was used, running all samples in triplicates according to our previously published protocol [76]. Cycling conditions of the RT-qPCR were as previously established [45,46,76]. RNase-free water served as the non-template control. The standard curves for target and reference genes were prepared by using pooled twofold-diluted series of cDNA (1:2-1:128) run as triplets to calculate PCR efficiencies (respective efficiencies given in Table 1). Analysis of the RT-qPCR results was performed using a modified model of the efficiency-corrected relative quantification according to Pfaffl [77], taking into account the expression of multiple reference genes for improved normalization [78]. As the model assumes stable expression of genes in the tissue, only *GAPDH* and *PTK2* (stable expression), but not *KDM4A* and *EIF4H* (unstable expression), were taken into consideration for normalization. Specific primer binding was confirmed by sequencing the PCR products (Microsynth AG).

### 4.5. Immunohistochemistry for OXTR and PGR and Evaluation of the Staining

For immunohistochemistry (IHC), sections of formalin-fixed, paraffin-embedded uterine tissue (3 µm) mounted on SuperFrost-Plus slides (Menzel Glaeser, Braunschweig, Germany) were treated following our previously published protocol [45,76]. After deparaffinization, rehydration, and antigen retrieval, blocking of unspecific binding sites was performed using 10% horse serum with 5% bovine serum albumin in ICC buffer for *OXTR* and 10% goat serum in PBS (ROTI^®^ Fair PBS 7.4, Carl Roth GmbH + Co. KG, Karlsruhe, Germany) for *PGR*, before samples were incubated with the respective first antibody (*OXTR*/*PGR*) overnight. All details about antibodies and dilutions are given in Table 2. Negative (buffer only) and isotype controls (Table 2) at the same protein concentration were included in each run. On the second day, the protocol differed considerably for detection of *OXTR* and *PGR*: for *OXTR*, a two-step horseradish peroxidase conjugated polymer system with DAB as a chromogen (SuperVision 2 HRP KIT PD000KIT; DCS Innovative Diagnostik-Systeme, Hamburg, Germany) was used according to the manufacturer’s instructions. For *PGR*, slides were incubated with the secondary antibody in 10% horse serum. Visualization of the immunopositive signal was performed with an immunoperoxidase system (VECTASTAIN PK-6100 ABC-Elite Standard: HRP and Vector Nova-RED Substrate Kit SK-4800; Vector Laboratories, Burlingame, CA, USA) according to the manufacturer’s instructions as previously described [76]. Then, all slides (*OXTR*/*PGR*) were counterstained with hematoxylin and dehydrated in increasing alcohol concentrations (70%, 96%, and 99%) and xylene before embedding with HistoKit (Assistant, Osterode, Germany). To avoid any bias due to differences between runs and to allow for better comparison, all samples were included in one run for *OXTR* and *PGR* each.

For evaluation of the staining, slides were visually examined under a light microscope (Olympus Bx 45, Olympus Europa SE & Co. KG, Hamburg, Germany) by two independent, blinded investigators. The localization of the immunopositive signals was evaluated descriptively. In addition, the staining intensity of the longitudinal and circular myometrial layer was scored independently and graded semi-quantitatively by both investigators, using an ordinal score system (1: weak, 2: moderate, and 3: strong staining). Results of the myometrial staining (mean of both investigators) were subjected to statistical comparisons (see below). In addition, the longitudinal and circular myometrial staining results of each dog were summarized as IP-Myoscore.

### 4.6. Statistical Analysis

Statistics was performed using Microsoft Excel 2016 (Microsoft Corporation, Redmond, WA, USA) and Graph Pad Prism9 software (GraphPad Software, Inc., La Jolla, CA, USA). The Shapiro–Wilk test was used to test for normal distribution. Data were presented as the arithmetic mean and standard deviation [x ± SD] when normally distributed. If not passing the test for normality, data were presented as the geometric mean and dispersion factor [xg (DF)]. Statistical differences were considered significant at a level of *p* ≤ 0.05.

The aim of the current research was to contribute to a better understanding of uterine inertia, by comparing PUI to OD data. Due to variable litter size in PUI, PUI-N was compared to OD, with litter size in OD being considered as normal according to earlier literature [45,74]. Datasets were separately evaluated for IP and UP. In addition, the impact of litter size was studied by comparison of PUI-S, PUI-N, and PUI-L in the IP samples.

To detect differences between P4 concentrations in PUI (*n* = 9) and OD (*n* = 7), an unpaired *t*-test was used as data were normally distributed.

For analysis of gene expression, most raw data were normally distributed. For the remaining (*OXTR*: PUI versus OD (UP); *PGR*: PUI-S versus PUI-N versus PUI-L), log transformation of data was performed revealing a normal distribution of log-transformed data according to the Shapiro–Wilk test. Subsequently, an unpaired *t*-test was used for comparison of PUI versus OD (IP/UP separately), PUI-N versus OD (IP only due to group size), and IP versus UP (summarizing PUI and OD) using the respective (raw/log-transformed) datasets. The influence of litter size (PUI-S versus PUI-N versus PUI-L) was studied using an ANOVA followed by Tukey’s multiple comparisons test if the ANOVA revealed *p* < 0.05. For a homogeneous data presentation and to ensure comparability, all gene expression results were presented as mean ± SD.

As a key role of *OXTR* and *PGR* in the myometrial layers was postulated in PUI, the immunohistochemical staining intensity was scored individually by two independent, blinded investigators for the longitudinal and circular myometrial layer. Calculation of Cohen’s kappa value [79] was used to assess inter-evaluator agreement for scoring of staining results. As the weighted K coefficient (κ = 0.89) revealed high inter-evaluator agreement, suggesting high comparability of the results of both investigators, the means of the semi-ordinal scores of both investigators were used for comparison of the individual myometrial layers (longitudinal/circular) between groups and for the different uterine localizations (i.e., PUI/OD IP/UP longitudinal versus PUI/OD IP/UP circular; PUI IP longitudinal/circular versus OD IP longitudinal/circular). Furthermore, all longitudinal myometrial staining results (PUI and OD summarized) were compared to all circular myometrial staining results (PUI and OD summarized) from IP and UP samples separately (PUI + OD IP longitudinal versus PUI + OD IP circular) and combining all layer-specific staining results from both localizations (all longitudinal IP + UP versus all circular IP + UP). In addition, the overall IP-Myoscore was calculated for each group and used for group-wise comparison (IP-Myoscore PUI versus IP-Myoscore OD). As none of the datasets were normally distributed following the Shapiro–Wilk test, either a Wilcoxon matched-pairs signed rank test for paired data or a Mann–Whitney test for unpaired data was carried out. Finally, the influence of litter size was studied as described above, comparing the separate layers and the IP-Myoscore using a Kruskal–Wallis test, followed by Dunn’s multiple comparisons test if *p* < 0.05.

## 5. Conclusions

As hypothesized, *OXTR* and *PGR* expression were altered in PUI. However, other than expected, both receptors were upregulated in PUI compared to OD. Whereas increased *PGR* expression might possibly potentiate the effect of the remaining circulating P4 by more effective binding, thereby inhibiting effective contractions, it remains to be clarified if upregulation of *OXTR* is a counterregulatory mechanism. In addition, the identified heterogeneity in *OXTR* protein expression between both myometrial layers, with a stronger staining in the circular myometrial layer, indicates functional differences that require further research for a better understanding of canine uterine contractility.

In conclusion, the numerous results of this and our other recent studies [15,16,17,45,46] give promising new clues to potential causes of PUI, but also indicate novel approaches for further research and contribute to a better understanding of myometrial contractility. Clarifying PUI is important not only to understand its underlying etiology but also to identify other treatment options in addition to surgery.

## Figures and Tables

**Figure 1 ijms-23-13601-f001:**
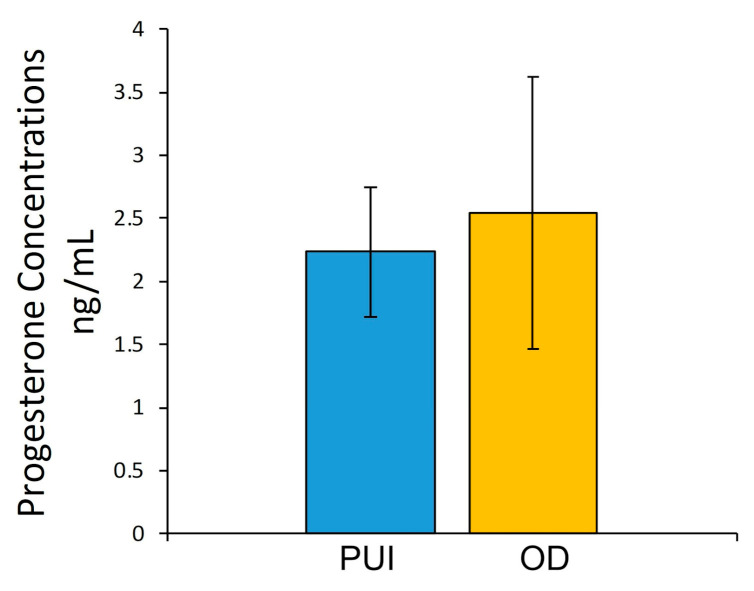
Serum P4 concentrations (ng/mL) immediately before C-section in bitches with primary uterine inertia (PUI) and obstructive dystocia (OD). Results presented as mean ± SD.

**Figure 2 ijms-23-13601-f002:**
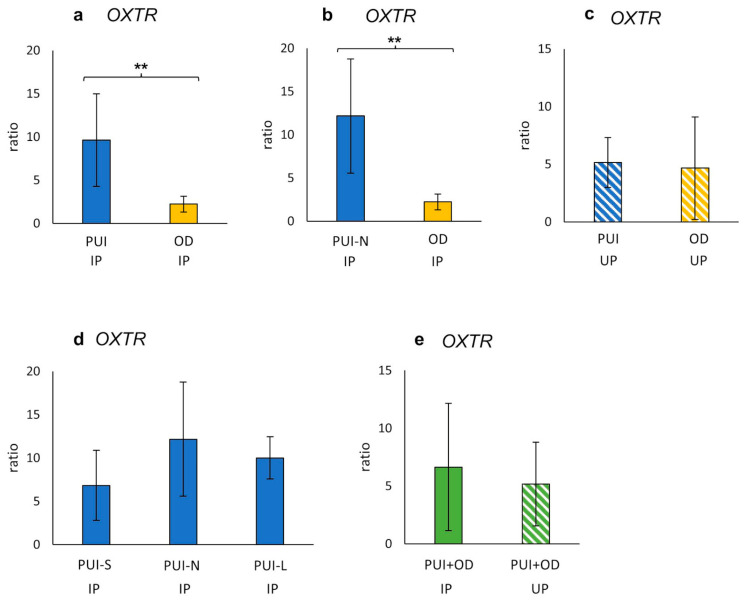
Interplacental (IP) (**a**,**b**,**d**) and uteroplacental (UP) (**c**) *OXTR* mRNA expression (ratio) in bitches diagnosed with (**a**,**c**) PUI (*n* = 11) or (**b**) PUI-N (*n* = 4), respectively, and OD (*n* = 8). (**d**) *OXTR* mRNA expression differentiating uterine inertia subgroups according to litter size (PUI-S: *n* = 4; PUI-N: *n* = 4; PUI-L: *n* = 3). (**e**) *OXTR* mRNA expression summarized for both groups (PUI + OD) for comparison of location (IP versus UP). Results presented as mean ± SD. Bars with asterisks differ significantly (** *p* < 0.01).

**Figure 3 ijms-23-13601-f003:**
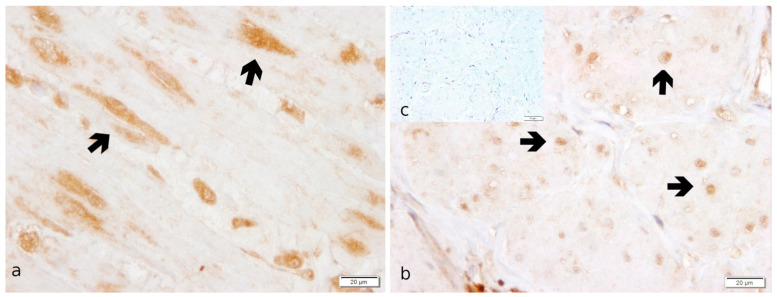
*OXTR* protein localization as revealed by IHC staining in canine uterine interplacental tissue. (**a**) myometrium, circular layer; (**b**) myometrium, longitudinal layer; (**c**) isotype control for *OXTR* given as inset. Independent of groups, myocytes (➔ black bold arrow) stained significantly stronger in the circular myometrial layer compared to the longitudinal one.

**Figure 4 ijms-23-13601-f004:**
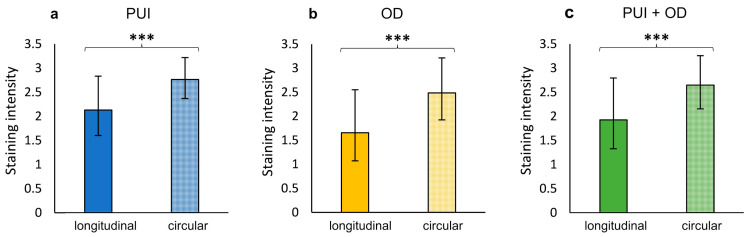
Comparison of *OXTR* IP staining intensity between myometrial layers (longitudinal/circular) for (**a**) PUI (*n* = 12) and (**b**) OD (*n* = 8), respectively, and (**c**) regardless of the group (PUI + OD). Results presented as mean ± SD. Bars with asterisks differ significantly (*** *p* < 0.001).

**Figure 5 ijms-23-13601-f005:**
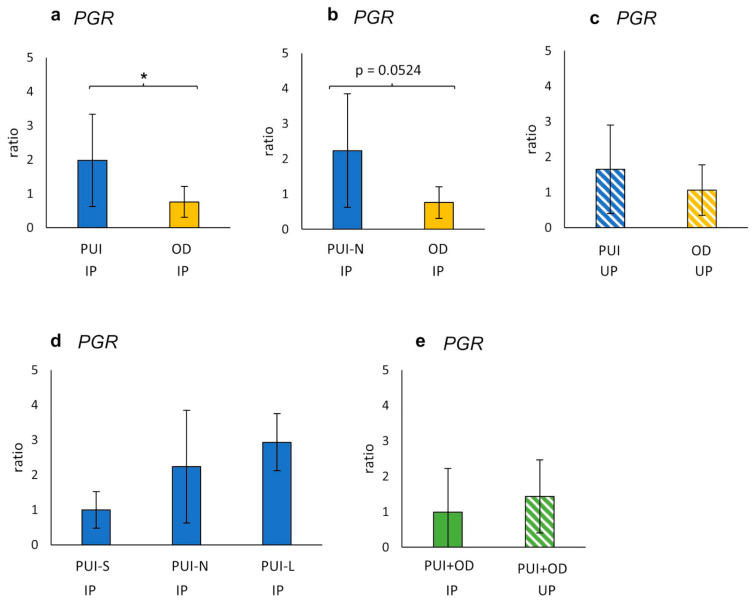
Interplacental (IP) (**a**,**b**,**d**) and uteroplacental (UP) (**c**) *PGR* mRNA-expression (ratio) comparing (**a**,**c**) PUI (IP *n* = 11; UP *n* = 4) or (**b**) PUI-N (IP only, *n* = 4), respectively, and OD (IP *n* = 8; UP *n* = 5). (**d**) *PGR* mRNA expression differentiating uterine inertia subgroups according to litter size (PUI-S: *n* = 4; PUI-N: *n* = 4; PUI-L: *n* = 3). (**e**) *PGR* mRNA expression summarized for both groups (PUI + OD) for comparison of location (IP versus UP). Results are shown as mean ± SD. Bars with asterisk differ significantly (* *p* < 0.05); a trend (*p* = 0.0524) is also given.

**Figure 6 ijms-23-13601-f006:**
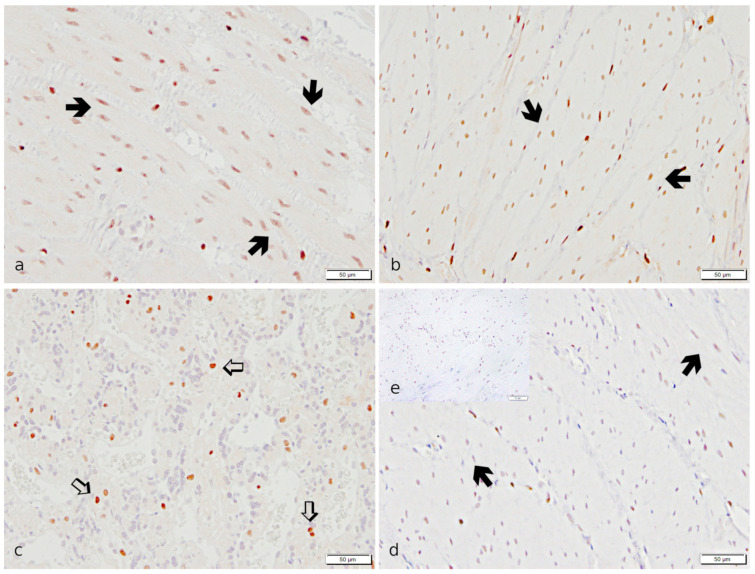
Specific immunostaining for *PGR* revealed in canine uterine tissue. (**a**,**b**,**d**): IP, (**c**,**e**): UP. (**a**,**b**,**d**,**e**): Myometrium; (**a**,**d**): Stratum circulare; (**b**): Stratum longitudinale; (**c**): Placenta. Strong immunopositive signals for *PGR* are visible in myocytes (➔, black bold arrow) of both myometrial layers and maternal decidual cells (⇨, white bold arrow) in the placenta in the PUI group (**a**–**c**). Myometrial staining appears weaker in OD (**d**) (representative image from the OD group). (**e**) The isotype control is devoid of signals (insert).

**Figure 7 ijms-23-13601-f007:**
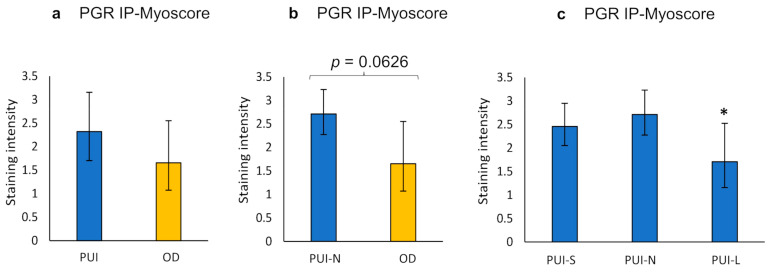
Interplacental myometrial staining score (IP-Myoscore) for *PGR* in PUI (*n* = 12), PUI-N (*n* = 4), and OD (*n* = 8) as obtained by immunohistochemistry. Comparison of IP-Myoscore of (**a**) PUI or (**b**) PUI-N, respectively, and OD and (**c**) within PUI subgroups according to litter size. Results are presented as geometric means with dispersion factor. Asterisks show significant differences (* *p* < 0.05).

**Table 1 ijms-23-13601-t001:** Sequence of primers for RT-PCR and RT-qPCR, amplicon length, efficiency, and accession number.

Primer	Accession Nr.	Forward Sequence (5′→3′)	Reverse Sequence (5′→3′)	Amplicon Length (bp)	Efficiency
*OXTR*	NM_001198659.1	GGATCACGCTCTCCGTCTACA	CGTCTTGAGTCGCAGGTTCTG	98	2.08
*PGR*	NM_001003074.1	CGAGTCATTACCTCAGAAGATTTGTTT	CTTCCATTGCCCTTTTAAAGAAGA	113	2.07
*PTK2*	XM_038685127.1	AGATGCTGACCGCTGCTCAT	TCAGTGTGGCCTCGTTGGTC	104	1.98
*GAPDH*	NM_001003142	GGCCAAGAGGGTCATCATCTC	GGGGCCGTCCACGGTCTTC	229	1.93
*EIF4H*	XM_038667880.1	GGAGTGTGCGGCTAGTCAGA	ACCCAACAGTGCACCATCGTA	199	1.97
*KDM4A*	XM_038687969.1	CCCGGCGGTGGATTGAGTAT	AACTCGGCTGCTTCTGGTGT	181	2.04

**Table 2 ijms-23-13601-t002:** Overview of primary and secondary antibodies used for IHC. Stock Keeping Unit (SKU) and producer given (* Abcam, Cambridge, UK; ** Thermo Fisher Scientific, Waltham, MA, USA; *** Vector Laboratories Burlingame, CA, USA).

Antibody	Source	Clone	Dilution (µg/µL)	SKU	Secondary Antibody	Isotype Control
*OXTR*	Rabbit	Monoclonal	0.02	ab217212 *	n.a. ^1^	Rabbit IgG *** (I-1000 Control Antibody)
*PGR*	Mouse	Monoclonal	0.02	PRAT 4.14 **	Horse anti-mouse *** (BA-2000)	Mouse IgG *** (I-2000 Control Antibody)

n.a. ^1^: not applicable as the SuperVision 2 HRP KIT PD000KIT was used.

## Data Availability

The data presented in this study are available in Appendix A of this paper. Further datasets are available on request from the corresponding author.

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
