# Peer review of "Involvement of Oxytocin and Progesterone Receptor Expression in the Etiology of Canine Uterine Inertia"

_ijms, 2022, doi:10.3390/ijms232113601_

Round 1
Reviewer 1 Report
The manuscript aims at comparing OXT receptors and PG receptors expressions in uterine tissue obtained during C-section due to primary uterine inertia and obstructive dystocia in dogs. The introduction and the whole manuscript are clear, well-written, and complete. The study design is appropriate and well-described, as well as the results and discussion. I think the paper is worthy of publication. I only have some minor revisions to ask. to authors.
General comments:
- I see that the lack of circulating oxytocin determination was acknowledged and explained by the authors (lines 246-250). These data would have allowed a better clarification regarding the altered mechanism in OXY during PUI; I would suggest adding this in case of future research on the subject, due to its importance.
- I would suggest adding, if possible, the analysis of the relationship between litter size and OD, as litter size may be related also to OD and therefore also to PGR expression (as a cause or also consequence!)
Specific comments:
Abstract: mean P4 is not lower in PUI compared to OD (p=0.053), as also reported in lines 286-287 of the discussion. There is only a tendency, so please correct
Introduction: delete lines 91-94
Discussion: delete line 346
materials and meth: delete lines 515-517
Reviewer 2 Report
I think that the subject of the work is of interest and that the topic of the manuscript is appropriate for the Journal. The information is of significant interest to the Journal's readers.
The title is nice, however, I suggest to change it as “Involvement of oxytocin and progesterone receptor expression in the etiology of canine uterine inertia”.
Please, avoid the use of personal form (e.g. our, we etc.) throughout the manuscript.
The abstract adequately summarize methodology, results, and significance of the study. However,
Authors should add some information on animals as age and they should indicate the statistical analysis applied on obtained data.
The introduction section is well written and it falls within the topic of the study.
Lines 91-94: I think that the sentence “2. Results. This section may be divided by subheadings. It should provide a concise and precise description of the experimental results, their interpretation, as well as the experimental conclusions that can be drawn.” should be removed from introduction section.
The section of Materials and Methods is clear for the reader and it meticulously describes the methods applied in the study. However, Authors should check this section and correct many punctuation errors. Moreover, Authors should indicate the inclusion criteria for animals enrolment.
Regarding statistical analysis, Authors wrote that they checked the normal distribution of data by a normality test. Please, report the results about this test.
The findings obtained in the study were well reported in the results section and the figures, which are generally good, well represent the results.
Results are well discussed and justified with appropriate references in Discussion section. However, Authors should make the Discussion section more harmonic and easy for the reader.
Conclusion section is clear and well written; indeed, Authors well summarize the major findings and they well emphasize the significance of the study.
Authors should check and standardize the references in the list according to journal guidelines.
Reviewer 3 Report
The idea of the study is easy and concise. I don't have any major concerns. The are some minor points which in my opinion missing and they would improve deeply value of presented manuscript.
You presented only morphological study results. Whereas in such cases we cannot distinguish between reason and result, we just have potential association without direction of relationship. That's obvious, however it should be stated clearly and explained to the reader.
I observe missing of important part of your experiment. There is no data about concentration of the OXT in serum. As a Ob/Gyn I realize difficulties in measurement and interpretation of such particule with short half life, but it would significantly improve study, especially in regard to analysis of P4 and their receptors... As you didn't collect the blood for subsequant studies at this time you cannot expand this experiment. Take it into account, if not not maybe in future.
Figures with IHC are deeply disappointing. As this is the only method in presented study images should be perfect. Contrast and magnification are bad. Fig 3c? Fig 6e? - useless. Counterstaining should be stronger.
References are well collected and properly presented.
After introduction of minor changes I would recommend for publication.
Author Response
Dear reviewer 3,
thank you very much for your nice and kind review.
Please find our comments and answers below - point-by-point.
We hope that with the respective changes the manuscript can be accepted for publication.
Sincerely,
Sandra Goericke-Pesch
I observe missing of important part of your experiment. There is no data about concentration of the OXT in serum. As a Ob/Gyn I realize difficulties in measurement and interpretation of such particule with short ha-lf life. but it would significantly improve study. especially in regard to analysis of P4 and their receptors... As you didn't collect the blood for subsequant studies at this time you cannot expand this experiment. Take it into account. if not not maybe in future.
Answer: Thank you very much for this suggestion. Despite the mentioned difficulties. the determination of oxytocin in serum would certainly be very informative and therefore the logical next step for further investigations.
Figures with IHC are deeply disappointing. As this is the only method in presented study images should be perfect. Contrast and magnification are bad. Fig 3c? Fig 6e? - useless. Counterstaining should be stronger.
Answer: The resolution of the original images is higher and of good quality, it might seem to be especially bad within the text. We agree that contrast of isotope controls had been bad why we improved the respective figures. We hope that you are satisfied now.